# Intracellular Metabolomics Switching Alters Extracellular Acid Production and Insoluble Phosphate Solubilization Behavior in *Penicillium oxalicum*

**DOI:** 10.3390/metabo10110441

**Published:** 2020-10-31

**Authors:** Yifan Jiang, Fei Ge, Feng Li, Dayi Zhang, Songqiang Deng, Jiang Tian

**Affiliations:** 1Department of Environment, College of Environment and Resources, Xiangtan University, Xiangtan 411105, China; jyfcjzx@163.com (Y.J.); gefei@xtu.edu.cn (F.G.); Lifeng6220@xtu.edu.cn (F.L.); 2School of Environment, Tsinghua University, Beijing 100084, China; zhangdayi@tsinghua.edu.cn; 3Research Institute for Environmental Innovation (Tsinghua-Suzhou), Suzhou 215009, China; izzydeng1987@163.com

**Keywords:** phosphorus sources, *Penicillium oxalicum*, phosphate solubilization, low-molecular-weight organic acids, metabolomics

## Abstract

This research aims to understand the precise intracellular metabolic processes of how microbes solubilize insoluble phosphorus (Insol-P) to increase bio-available P. Newly isolated *Penicillium oxalicum* PSF-4 exhibited outstanding tricalcium phosphate (TP) and iron phosphate (IP) solubilization performance—as manifested by microbial growth and the secretion of low-molecular-weight organic acids (LMWOAs). Untargeted metabolomics approach was employed to assess the metabolic alterations of 73 intracellular metabolites induced by TP and IP compared with soluble KH_2_PO_4_ in *P. oxalicum*. Based on the changes of intracellular metabolites, it was concluded that (i) the enhanced intracellular glyoxylate and carbohydrate metabolisms increased the extracellular LMWOAs production; (ii) the exposure of Insol-P poses potential effects to *P. oxalicum* in destructing essential cellular functions, affecting microbial growth, and disrupting amino acid, lipid, and nucleotide metabolisms; and (iii) the intracellular amino acid utilization played a significant role to stimulate microbial growth and the extracellular LMWOAs biosynthesis.

## 1. Introduction

As a structural and functional component, phosphorus (P) play critical roles in supporting global food requirements and maintaining the vitality of agricultural organisms on Earth [1]. Also, P is an indispensable element for plant growth and an important component of agricultural production. However, the use of soil P still poses a major challenge for plants, as P is non-renewable, and PO_4_^3^^−^ is strongly absorbed as Ca, Al, and Fe (hydro) oxides [2,3]. Indeed, less than 0.1% of total soil P exists in bioavailable form for plant growth and crop use [4]. Phosphate-solubilizing microorganisms (PSM) are key regulators in the transformation and biogeochemical cycles of soil P by enhancing the bioavailability of P, HPO_4_^2^^−^, and H_2_PO_4_^−^ through inorganic P solubilization and organic P mineralization. Many previous studies have used PSM in agricultural systems for their eco-friendly and simple ability to liberate P from organic and inorganic resources to make it bioavailable [5,6,7,8].

It was reported that inorganic P solubilization by PSM relies primarily on H^+^ extrusion to lower the pH and the production of low-molecular-weight organic acids (LMWOAs), such as gluconic, acetic, malic, succinic, formic, oxalic, and citric acids [9]. The hydroxyl and carboxyl groups of LMWOAs chelated Ca^2+^, Al^3+^_,_ and Fe^3+^ cations, promoting heterotrophic PSM to release soluble P by dissolving inorganic P [10]. Accordingly, the effectiveness of P solubilization depends on the LMWOAs and their molecular structure, such as the number of carboxyl groups and other carboxyl or hydroxyl moieties [11,12]. Efthymious et al. reported that tricarboxylic and dicarboxylic acids could chelate with P more strongly than monocarboxylic acids [4]. However, LMWOAs production was dependent on insoluble P sources [13]. For instance, gluconic, oxalic, and malic acids were produced by *Streptomyces* spp. in the medium supplemented with Ca_3_(PO_4_)_2_, while succinic and formic acids were detected when the strains grew in the medium supplemented with FePO_4_ [14]. Clearly, PSM secreted different LMWOAs according to the characteristic of P sources, which complicates the mechanism of P solubilization.

Along with the dependence of inorganic P solubilization on LMWOAs production, changes in P sources may also influence P solubilization performance by altering the intracellular metabolism of PSM. Extracellular organic acids, primarily LMWOAs, are the final products of intracellular metabolites [15]. Thus, research on changes in the metabolite profile may be useful for understanding LMWOAs metabolism during P solubilization. Metabonomics, based on the construction and modeling of metabolic networks, could be used to understand the characteristics of intracellular metabolic mechanisms from a holistic perspective [16,17]. Zhao et al. found that the elevated extracellular protein-to-polysaccharides ratio resulted in the obviously increased extracellular hydrophobicity and aggregation capacity through the intracellular metabonomic analysis of microbial community [18]. Other studies showed that PSM *Agrobacterium* sp. Could enhance the Cd mobilization by intracellular succinic, fumaric acid production, and amino acid biosynthesis [19]. However, conclusive evidence for the intracellular metabolic regulation and extracellular LMWOAs production in PSM in response to different P sources is still insufficient.

Here, we compared the P-solubilizing fungus (PSF) *Penicillium oxalicum* PSF-4 (*P. oxalicum*) with tricalcium phosphate (Ca_3_(PO_4_)_2_, TP) and iron phosphate (FePO_4_, IP,) solubilization activities. The primary aims of this study were to a) investigated the physiological changes (including biomass and extracellular acids) of *P. oxalicum* at different P sources; b) assessed the core metabolic pathways in response to P sources in *P. oxalicum* using liquid chromatography-mass spectrometry-based metabolomics; c) established the relationship between intracellular metabonomics and extracellular LMWOAs production for inorganic P solubilize in response to the environmental P source. These results will provide new insight into the intracellular metabolomics switching alters extracellular LMWOAs production and P solubilization behavior in *Penicillium oxalicum*, which improve its agricultural application potentials in soil P regulations.

## 2. Results

### 2.1. P-Solubilization Performance and Microbial Growth of P. oxalicum

The maximal concentrations of soluble-phosphorus (P_sol_) in the tricalcium phosphate (TP), iron phosphate (IP), and dipotassium phosphate (C) treatments during 13 days of culture were 1098.9, 31.3, and 1541.1 mg/L, respectively (Figure 1). As shown in Figure 1, pH decreased from 4.3 to 2.3 in TP treatment. Nevertheless, there were no significant changes in pH compared with the control and IP treatments during 13 days of culture (*p* < 0.05). P_sol_ concentrations in IP were less than 40 mg/L during 13 days of culture, which were almost 0.03-fold of that in TP. The maximal microbial biomass of *P. oxalicum* cultured for three days under TP, IP, and C reached 0.26, 0.22, and 0.17 g, respectively (Appendix A).

### 2.2. Extracellular Acid Production

Although oxalic, citric, gluconic, formic, malic, lactic, and acetic acids were detected in all treatments, the yield of low-molecular-weight organic acids (LMWOAs) in *P. oxalicum* was remarkably different across treatments. (Figure 2). The concentrations of total LMWOAs in TP peaked at 200.7 mg/L on day 3 (Appendix A), including acetic acid (33.9%, 68 mg/L), oxalic acid (27.4%, 55 mg/L), gluconic acid (23.8%, 47.8 mg/L) and fumaric or pyruvic acid (14.9%, Figure 2). However, total LMWOAs concentration in the IP treatment reached 1786 mg/L (Appendix A), and the predominant components were malic acid (49.8%, 890 mg/L), gluconic acid (34.5%, 617 mg/L) and oxalic acid (15.6%, 279 mg/L). In the C treatment, similar LMWOAs profiles were obtained as in the TP treatment, peaking on day 3 at 601.1 mg/L (Appendix A); however, the main LMWOAs were tartaric acid (314 mg/L), acetic acid (194 mg/L) and gluconic acid (92 mg/L). Across the three treatments, more types of LMWOAs were found in TP treatments compared with other treatments.

### 2.3. Metabolite Profiling of Intracellular Metabolites

A total of 3525 and 7356 non-targeted peaks were detected by UHPLC-Q-TOF in ESI positive and negative mode, respectively (Appendix A). QC samples had high repeatability and stability throughout the detection process (Appendix A), indicating that raw data and metabolic biomarkers representing intracellular metabolic differences were stable and reliable. Furthermore, one-way ANOVA and Tukey’s HSD posthoc analyses at *p* = 0.05 revealed that a total of 73 metabolites were integrated and isolated (Figure 3, Appendix A). A heat map showed that these metabolites with significant changes in relative abundance after three days of culture (*p* < 0.05) across the TP, IP, and C treatments (Figure 3A), which included oxalate, citrate, fructose, and glucose (Figure 3C). Most metabolites were at extremely low concentrations in the C treatment, except for some low-molecular-weight metabolites, such as oxalate, ribulose, and linoleic acids (Figure 3A). PCA score plots revealed differences in metabolite profiles and suggested that between-group variance could be easily resolved in TP, IP, and C (Appendix A). Additionally, discriminant analysis with the partial least square method (PLS-DA) and hierarchical clustering analysis (HCA) analyses were conducted to provide insight into specific differences in the metabolite profiles between treatments, reflecting the strong impacts of intracellular metabolic fold changes (FC) in response to different P sources (Appendix A).

### 2.4. Metabolic Response of P. oxalicum to P Sources

The 73 metabolites were then categorized into soluble P (Sol-P, C treatment) and insoluble P (Insol-P, including TP and IP treatments). The heat map revealed clusters of nine classes of compounds: LMWOAs, N-compounds, sugars, alcohols and polyols, phosphate, nucleosides, polyhydroxy and carbonyl, amino acids, and vitamins (Appendix A). Generally, the concentrations of amino acids, nucleosides, and phosphate in Insol-P treatments were dramatically reduced, while the LMWOAs and sugars were increased.

Important metabolites were selected from the volcano plot and the OPLS-DA loading S-plot (Appendix A). Based on the Kyoto Encyclopedia of Genes and Genomes pathway libraries, 14 intracellular metabolic pathways were up-regulated with Insol-P treatment *(p* < 0.05), involving amino sugar and nucleotide sugar, glyoxylate and dicarboxylate, lysine degradation and biosynthesis, and the TCA cycle (Appendix A). Detailed differences in the metabolites suggest that the levels of central metabolism differed markedly between the Insol-P and Sol-P treatments in *P. oxalicum* (Figure 4). Specifically, sugar and sugar alcohols (mannose, glucose, fructose, tagatose, galactinol, and glyceric), LMWOAs (citrate, malate, aconitic, aminoadipic, and propionic acid), amino acids (lysine, valine, methionine, leucine, arginine, tyrosine, serine, uridine, glutamine, and saccharopine) and phosphates (phosphorylcholine, glycerol 3-phosphoethanolamine, and adenosine monophosphate) were at relatively higher concentrations in *P. oxalicum* fermented with Insol-P. In contrast, only oxalic acid derived from the glyoxylate and dicarboxylate pathway, as well as lactic acid derived from pyruvate metabolism, were at higher concentrations in the Sol-P control. Gluconic, citric, succinic, and malic acids had similar FC increments in intracellular metabolism compared with extracellular acid production, while lactic and oxalic acid showed the opposite pattern (Figure 2 and Figure 4).

### 2.5. Metabolic Output Affected by Both Insol-P and Sol-P Sources

To further assess the metabolic effects of the P sources, we compared the metabolites in *P. oxalicum* treated with Insol-P and Sol-P to isolate biomarkers metabolites. A total of 42 metabolites with significant intensity increases (FC > 2 fold, *p* < 0.05) and decreases (log_2_ (FC) < −1, *p* < 0.05) were screened by paired analysis (Figure 4). Subsequently, the distribution of the regulated metabolites in intracellular metabolic pathways was further summarized, as shown in Figure 5 and Appendix A. Down-regulation of *P. oxalicum* in important intracellular metabolite pathways, such as the TCA cycle, carbohydrate metabolism, and purine metabolism, under culture with IP was observed by the down-regulation of acids, sugars, and nucleosides. Nevertheless, no significant changes were found in all of the detected amino acids by culturing with IP or TP in *P. oxalicum* (Figure 4 and Figure 6). In addition to amino acids, metabolites (including glyceric acid, malic acid, oxalate, lactate, ribulose, linolenic acid, UDP-N-acetylglucosamine, and uracil) were up- or down-regulated in all three treatments.

In terms of TP vs. Sol-P and IP vs. Sol-P, 45 and 55 metabolites with significant intensity increases and decreases were screened, respectively. Up-regulations of *P. oxalicum* protein, glutamate, glycerophospholipid metabolism, and the TCA cycle were observed based on the distribution of the regulated marker metabolites in TP treatment (Figure 4 and Figure 5). Additional, interesting patterns of regulation of D-fructose, D-glucose, and D-tagatose in four comparisons were observed in carbohydrate metabolism, in which only up-regulation was found in IP treatment, suggesting that the addition of IP has an effect on intracellular carbohydrate metabolites compared with TP and Insol-P. Moreover, as the results of extracellular LMWOAs, the concentration of gluconic acid in IP treatment was much higher than that in TP and C treatments. Similar regulatory patterns were also observed in the TCA cycle (for citrate and aconitic acid), and purine metabolism (for adenosine monophosphate), and only up-regulation was found in Insol-P treatments. Furthermore, glycerophospholipid metabolism and amino acid biosynthesis were more active under TP than under IP, and the lower concentrations of amino acids observed under IP might be utilized for carboxylic acid and glutathione metabolism.

## 3. Discussion

### 3.1. Effect of Insol-P Sources on Extracellular Acid Production during P Solubilization

During P solubilization, heterotrophic metabolism in PSM under low pH medium by H^+^ extrusion or LMWOAs secretion primarily relies on exogenous carbon sources [9]. However, the solubilization mechanism and altered acid production induced by insoluble P sources remain poorly understood [14,20]. Under Insol-P conditions, P-solubilization performance varies across strains of PSM and heterogeneous P sources [21,22]. Correspondingly, the present study demonstrated that the P-solubilization abilities in *P. oxalicum* followed the rank order of TP > IP, which can be explained by the difference in LMWOAs production (e.g., gluconic acid, oxalic acid citric acid) under different Insol-P sources [22]. In the TP treatment, various LMWOAs was detected (e.g., gluconic acid, oxalic acid, acetic acid, fumaric acid, and pyruvic acid). However, in the IP treatment, only malic and gluconic acids were detected. Previous work shows that higher amounts of gluconic acid (monocarboxylic acid with low P-chelating ability) are not directly correlated with higher degrees of P solubilization [23]. But it is certain that these LMWOAs secreted by *P. oxalicum*, such as oxalic acid, gluconic acid, and acetic acid, etc., will affect the Insol-P solubilization, due to different acidity coefficients. The acidity coefficient of LMWOAs reflects the ability of the acid to ionize H^+^ [24]. As shown in Appendix A, the comprehensive acidity coefficient of the TP group is lower than that of IP, as the acidity coefficients of fumaric (p*K*_a_ = 2.18) and pyruvic acids (p*K*_a_ = 2.39) are lower than malic (p*K*_a_ = 3.46) and gluconic acids (p*K*_a_ = 3.86) [24]. The lower acidity coefficient leads to higher extracellular ionized H^+^, which further improved the P_sol_ concentrations in TP treatment. These findings are consistent with the fact that P solubilization was affected by both pH and LMWOAs [25].

### 3.2. Intracellular Metabolic Pathways Responsible for Extracellular Acid Production

Since the biosynthesis of LMWOAs responsible for P solubilization in PSM is controlled by exogenous P sources, the intracellular metabolomics analysis related to the production of extracellular organic acids is expected to explain the P-solubilization mechanisms of PSM to different P sources. Central carbon metabolism pathways, such as the TCA cycle and glyoxylate and carbohydrate metabolism, were up- or down-regulated based on the levels of LMWOAs, such as citric, aconitic, and malic acid [26]. Consequently, the carbons used to synthesize acids were translocated to produce extracellular LMWOAs instead of microbial biomass [27]. In this study, a significant down-regulation of D-glucose, and D-fructose in carbohydrate metabolism was observed between IP vs. TP and IP vs. Sol-P, a pattern that was consistent with the higher concentrations of extracellular gluconic acid in the IP treatment (Figure 2 and Figure 5). In addition to gluconic acid, the presence of malic and citric acids was significantly increased by exposure to TP and IP (*p* < 0.05), indicating that the TCA cycle connecting glucose and glyoxylate metabolisms could be considered the hub for extracellular LMWOAs metabolic pathways. The down-regulation of citric acid, oxalic acid, D-glucose, and D-fructose in response to IP reflects the sensitivity of carbohydrate and glyoxylate metabolism to Insol-P stress, which resulted in a reduced accumulation of LMWOAs in cells compared with other treatments.

### 3.3. Regulation of Cellular Functional Metabolites and Production of Biomass

The effect of good P-solubilization performance under Insol-P coincided with the sustained growth of *P. oxalicum* biomass in TP and IP (Appendix A). Solans reported that the limitation in biomass production could stem from the stress-induced by Insol-P (TP < IP) during P solubilization [14]. Similarly, our data showed that the exposure of Insol-P poses potential effects to *P. oxalicum*, such as destructing the essential cellular functions, inhibiting the microbial growth, and disrupting the biological metabolism, including amino acid, lipid, and nucleotide metabolism (Figure 5 and Appendix A).

#### 3.3.1. Purine and Nucleotide Metabolism: Biomass Production

For most microorganisms, the intensity of intracellular nucleotide metabolism reflects the growth of organelles and the production of biomass [28,29]. The down-regulated adenine, adenosine monophosphate (AMP), and guanosine monophosphate indicated that nucleotide biosynthesis was sharply decreased in IP compared with TP (Figure 4), which is consistent with the previous study that has demonstrated that the effect of the stress-induced by different plenty of Insol-P on biomass production can be characterized by antagonistic or synergistic interactions [27]. Similarly, the biomass in this study as followed by TP, IP, and Sol-P, one reasonable explanation for this result is that compared with the IP and Sol-P treatments, the TP treatment stimulated massive nucleotide synthesis and enhanced intracellular signals enzymes activity that triggered physiological changes in response to environmental stimuli [30,31]. As showed in Figure 4, significantly increased AMP production was observed in the TP treatment compared with the IP and Sol-P treatments. Indeed, large amounts of AMP are usually detected when energy molecules have been dephosphorylated to support fundamental cellular functions [32]. Thus, substantially higher levels of cellular nucleotide synthesis provided molecular evidence suggesting that biomass growth was more rapid under TP stress [33].

#### 3.3.2. Phospholipid Metabolism: Lipid Biosynthesis Form Cellular Membranes

Unlike purine metabolism and nucleotide synthesis, IP, and TP exposure showed similar levels of regulation with respect to protein and phospholipid metabolism. The metabolites involved in phosphatidylcholine synthesis (acetylcholine and choline) were balanced based on the levels of Sol-P control, but phosphorylcholine and sn-glycerol 3-phosphoethanolamine (sn-glycerol 3-PEA) were significantly increased (FC values > 2), especially under TP exposure—and this appeared to be related to Sol-P control (Figure 4 and Appendix A). One reasonable explanation for this pattern is that all of the metabolites involved in lipid biosynthesis to form cellular membranes, the observed biomass switches (TP > IP > Sol-P in Appendix A) were consistent with the phospholipid metabolites that were observed to be released in all of the treatments [34]. Therefore, to a large extent, the up-regulation of phospholipid metabolism could promote the formation of the membrane and reflect the production of biomass eventually. Since only a few biomarker metabolites have been detected, we speculate that variation in the phospholipid metabolic pathway might not have the significant effect on biomass production [28,33].

#### 3.3.3. Amino Acid Production: Energy Production and Biomass Yield

The most notable functional metabolite changes were observed in amino acid production between Insol-P and Sol-P exposure. Nevertheless, none of the detected amino acids was found to be up- or down-regulated between IP and TP exposure (Figure 4 and Appendix A). The presence of high levels of amino acids is thought to contribute to the increased degradation or breakdown of proteins for the synthesis of new proteins for survival during environmental stress [35]. In this study, higher levels of all amino acids were observed under the stress of P deficiency through culture with insoluble TP and IP, while the amino acid synthesis pathways, including aminoacyl-tRNA biosynthesis and amino sugar, lysine, and glutamate metabolism, were up-regulated (Figure 6 and Appendix A). In addition, the fact that protein synthesis is a high-energy process (consuming 50–60% of total ATP to polymerize amino acids), which resulted in significant (*p* < 0.05) higher levels of energy production and biomass yield in Insol-P treatments (Appendix A) [33].

### 3.4. LMWOAs Biosynthesis through Amino Acid Utilization

The levels of all of the detected amino acids increased during both TP and IP-solubilizing processes, and no significant differences were observed between IP and TP (Figure 4 and Figure 6, and Appendix A). This interesting observation indicated that P-deficient conditions with Insoluble-P caused *P. oxalicum* to utilize amino acids with different strategies, including biomass yield, energy production, and LMWOAs biosynthesis.

#### 3.4.1. Glucogenic Amino Acids

Amino acids were synthesized through amino sugars, glutamate (alanine, aspartate, and glutamate), and lysine metabolism with exogenous glucose or protein degradation [35]. However, higher levels of glucogenic amino acids (including serine, arginine, glutamate, glutamine, methionine, tyrosine, and valine) were detected compared with ketogenic amino acids (leucine and lysine) in both the TP and IP treatments (Appendix A). The increase in glucogenic amino acids may stem from the higher production of glucose via gluconeogenesis [36], and is consistent with the idea that a glucogenic amino acid-based feeding strategy would induce a higher release of glucose [37], which might also explain the markedly increased level of extracellular gluconic acid observed in insoluble TP and IP samples (Figure 2 and Figure 4).

#### 3.4.2. Metabolism Associated with Propionic Acid

Propionic acid is another important metabolic product that is responsible for amino acid utilization and LMWOAs production under P-deficiency stress. This short-chain saturated triglyceride is a major product of amino acid and dicarboxylic acid metabolism (consuming glycerol as an ideal source) and a precursor of carboxylic acid metabolism [36,38]. Microbes can convert propionic acid to propionyl-CoA and utilize it for carboxylic acid (including gluconic, galactaric, and glucuronic acid) production [36,39]. Significantly increased levels of propionic acid were only observed in IP treatment (Figure 4 and Figure 6), suggesting that amino acids were used and converted into gluconic acid in the IP treatment, which was consistent with the lack of a significant increase in the amino acids methionine, serine and tyrosine involved in propionic acid production and the increase in glyceric acid (FC > 2) in IP treatments (Figure 4) [40]. Meanwhile, the fact that the level of extracellular gluconic acid production was highest in the IP treatment (Figure 2), which could be explained by the IP-induced up-regulated carbohydrate metabolism as indicated by the increase in amino acid utilization and propionic acid production. Overall, our findings highlight the significant role that amino acid utilization plays in releasing important metabolites to stimulate *P. oxalicum* microbial growth and the extracellular LMWOAs biosynthesis in response to P sources.

## 4. Materials and Methods

### 4.1. Biological Materials and Growth Conditions

Soil samples were collected from an abandoned chemical factory that previously produced phosphate rocks in Hunan Province, China (29°30′41″ N, 113°15′28″ E) using sterilized forceps and were stored in sterilized plastic bags. After the samples were air-dried, 1.0 g of soil was suspended in 9.0 mL of phosphate buffer saline (PBS, pH 7.2), serially diluted (1:10), and spread on Pikovskaya’s (PKV) agar containing TP as the P source [41]. Plates were incubated at 28 °C for 7 days, and positive colonies showing a clear halo were selected as PSM (Appendix A). Among the PSM, one of the efficient isolates of *Penicillium* strain PSF-4 was identified based on nucleotide sequence data from the ITS of rRNA. The ITS of rRNA fragments were amplified using the primer pair of ITS1 (5′-TCCGTAGGTGAACCTGCGG-3′) and ITS4 (5′-TCCTCCGCTTATTGATATGC-3′) [42]. Nucleotide sequence data were compared with GenBank data (http://www.ncbi.nlm.nih.gov/) using the BlastN search (Appendix A), and PSF-4 was stored in China General Microbiological Culture Collection Center (No. M2019233). *P. oxalicum* was first cultured in 100 mL of PKV broth containing 50 g/L of TP and incubated at 28 °C on a rotary shaker (150 rpm/min) for 7 days, and 100 mL of supernatant was transferred to PDA (Potato Dextrose Agar) broth medium for 48 h.

### 4.2. P-Solubilizing Experiments and Chemical Determination

After enrichment culturing in PDA broth medium for 48 h, 10 μL of *P. oxalicum* spore suspension (10^7^ cfu/mL) was centrifuged, washed, and then added in 50 mL of PKV broth containing 5 g/L TP (*K*_sp_ = 2.07 × 10^−33^), IP (*K*_sp_ = 1.3 × 10^−22^) or soluble KH_2_PO_4_ (control group, C) as the sole P source, respectively [43]. Approximately 50 μL of medium suspension was taken every 24 h for 13 days and centrifuged at 12,000 rpm for 10 min. The concentration of soluble-phosphorus (P_sol_) in the supernatant was estimated by colorimetry [44], and the pH of the supernatant was measured using a pH meter (PHS-3C pH Meter, REX). The microbial biomass in the cell cultures in each treatment was harvested on day 3 and calculated following filtering the culture medium, edulcorated by PBS, and finally, air-drying.

### 4.3. Extraction and Identification of Extracellular Acids

The concentrations of LMWOAs were measured by high-performance liquid chromatography (HPLC). Briefly, 2 mL of suspension after 3 days of culture was centrifuged at 12,000 rpm for 2 min, suspended, and passed through a 0.22 μM PTFE membrane filter. The samples were then analyzed using an equipped with a variable wavelength HPLC (Agilent 1260 Infinity, Santa Clara, CA, USA) detector G1314F and a Platisil ODS C_18_ column (4.6 mm × 250 mm, 5 μm). Details on the mobile phase and the calculation process are provided in the Appendix A.

### 4.4. Intracellular Metabolite Extraction and Quality Control

After three days (the day where pH and P_sol_ were tremendously changed) of culture in PKV medium supplemented with TP, IP, and C, globular mycelia of each treatment were collected, and the rapid filtration step was repeated for eight times, followed by NaCl washing to remove the extracellular metabolites [45]. The biomass samples were quickly harvested and went through two cycles of quenching using 800 μL of cold methanol/acetonitrile (1:1, (*v/v*)) at −80 °C by 60 s of vortex shocking and 30 min of cryogenic ultrasonic extraction. The supernatants were then harvested after precipitation of cellular proteins at −80 °C for 60 min and centrifugation of lysates at 14,000 *g* for 20 min. All of the supernatants were completely lyophilized, stored at −80 °C and prepared so that the intracellular compounds could be determined. In addition, quality control (QC) samples were conducted to confirm the efficacy of the experiments. Details are provided in the Appendix A.

### 4.5. Statistical Analysis

The phenotypic data were conducted using GraphPad Prism 7.0 (Graph pad Software, San Diego, CA, USA) and SPSS 22.0 (Inc., Chicago, IL, USA). One-way and two-way analysis of variance (ANOVA) with Tukey’s multiple comparison post hoc tests were used to compare the difference of P_sol_ concentrations, pH, and LMWOAs production between TP, IP, and C treated group. For metabolite data, we used an integrity test followed by the deletion of 50% of the missing metabolites and the extremes in a group, then applied pareto-scaling to pre-analysis. Multidimensional statistical analysis, including unsupervised principal component analysis (PCA), discriminant analysis with the partial least square method (PLS-DA), discriminant analysis of orthogonal partial least squares (OPLS-DA) and HCA were performed following generalized log transformation of raw data with MetaboAnalyst 3.0 (GenomeCanada, Ottawa, Canada) [46]. Seven-fold cross-validation and response permutation tests were used to evaluate the robustness of the model. The variable importance in the projection (VIP) value of each variable in the OPLS-DA model was calculated to indicate the contribution of each variable to the classification. Furthermore, univariate student’s t-tests were used to measure the significance of metabolites. VIP values > 1 and *p* < 0.05 was considered to be significant. Volcano plots of fold-change and correlation heatmaps were performed with R, log-transformation, and hierarchical clustering, and metabolic pathways were performed with MetaboAnalyst 3.0 [18].

## 5. Conclusions

PSM were key regulators in the biogeochemical cycles of environmental P by enhancing the P bioavailability through inorganic P solubilization. Using intracellular metabolomics, our study provided novel illustrations into the LMWOAs production and P-solubilization performance of *P. oxalicum* PSF-4 in response to different P sources. The entire metabolic process with significantly different metabolites revealed that environmental P sources could cause differences in the production of extracellular LMWOAs and microbial biomass, as was indicated by different levels of intracellular carbohydrate, nucleoside, phospholipid, and amino acid metabolism in *P. oxalicum* PSF-4. Therefore, the interaction network diagrams synthetically illustrate the diversities of metabolic pathways in response to P sources, providing new molecular insight into the P solubilization mechanism and environmental applications of *P. oxalicum*. For comprehensive and specific P solubilizing mechanism, multiple highly representative strains are needed for further studies.

## Figures and Tables

**Figure 1 metabolites-10-00441-f001:**
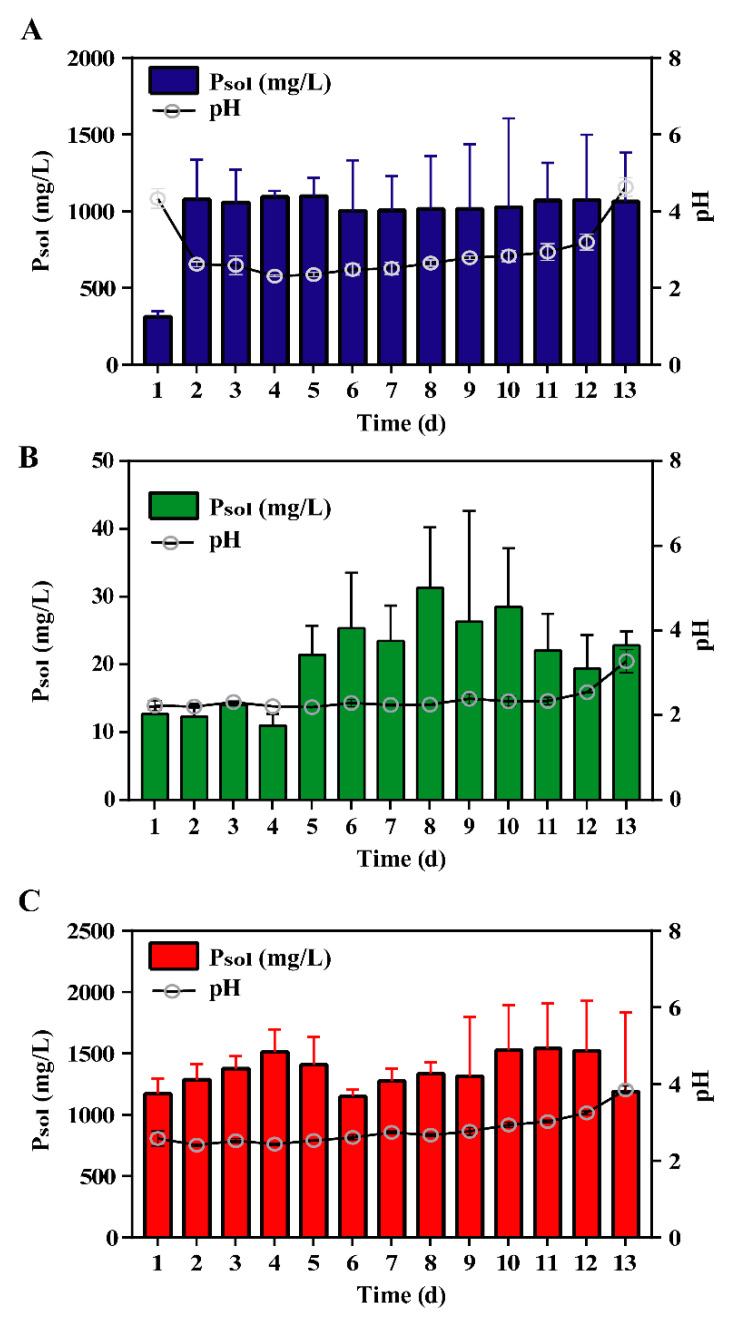
Changes of soluble-phosphorus (P_sol_) by *Penicillium*
*oxalicum* with (**A**) tricalcium phosphate, (**B**) iron phosphate, and (**C**) dipotassium phosphate as the sole P source during 13 days of culture. Data are the average result of three replicates with standard deviations.

**Figure 2 metabolites-10-00441-f002:**
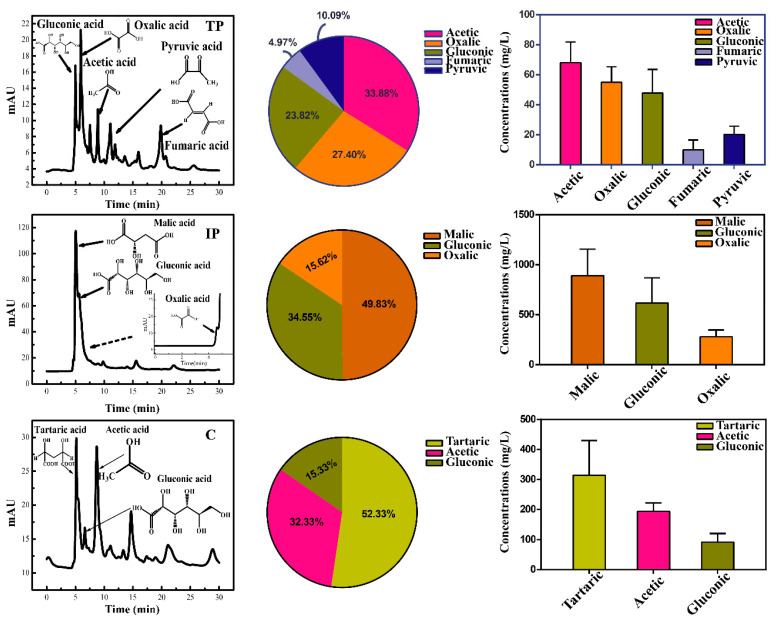
High-performance liquid chromatography (HPLC) diagrams for low-molecular-weight organic acids (LMWOAs) secreted by *P. oxalicum* cultivated with tricalcium phosphate (TP), iron phosphate (IP), and dipotassium phosphate (C) after three days of cultivation during P solubilization.

**Figure 3 metabolites-10-00441-f003:**
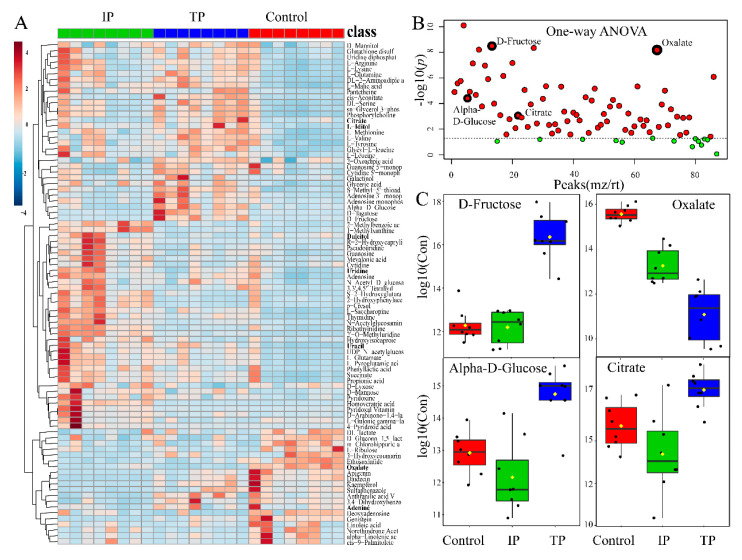
Metabolic analysis in *P. oxalicum* with tricalcium phosphate (TP), iron phosphate (IP), and dipotassium phosphate (C) exposure. (**A**) Heat map visualization, hierarchical clustering analysis (HCA) provided intuitive visualization of all 73 identified metabolites with generalized logarithm transformation of concentration. Relative metabolite levels of three treatments were performed using an average clustering algorithm and Pearson’s correlation distance measure. (**B**) Important features selected by one-way ANOVA and Tukey’s HSD with *p*-value threshold 0.05 (Appendix A). Data represent −log_10_ (raw *p*-value) changes. The dotted line represents the threshold *p* = 0.05. (**C**) Boxplot of 4 selected metabolomics features from the one-way ANOVA. The black dots represent the −log_10_(raw concentrations) from all the eight samples, and the notch indicates the 95% confidence interval around the median of each treatment. The mean concentration of each treatment is indicated with a yellow diamond.

**Figure 4 metabolites-10-00441-f004:**
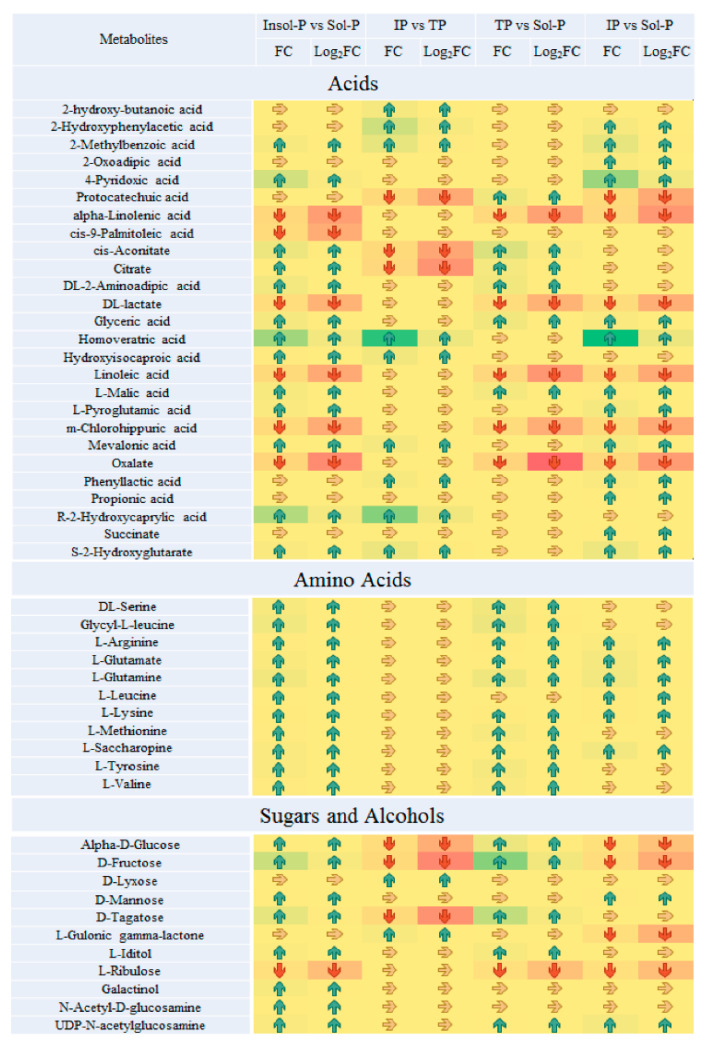
Paired fold change analysis of Important metabolites that are affected by P sources (insoluble P (Insol-P) and soluble P (Sol-P), tricalcium phosphate (TP) vs. iron phosphate (IP), TP vs. Sol-P and IP vs. Sol-P). TP refers to the tricalcium phosphate group; IP refers to the iron phosphate group; Insol-P refers to insoluble P, including TP and IP groups; Sol-P refers to soluble P (dipotassium phosphate) group. Arrows of average fold changes (FC) and log_2_(FC) are listed and painted with increased coded “blue” (FC > 2, log_2_(FC) > 1) and decreases coded “red” (FC < 0.5, log_2_(FC) < −1) with significant changes (*p* < 0.05). The color of arrow listed “yellow” represent the undetected metabolites or metabolites with no significant FC values changes (0.5 < FC < 2, *p* < 0.05).

**Figure 5 metabolites-10-00441-f005:**
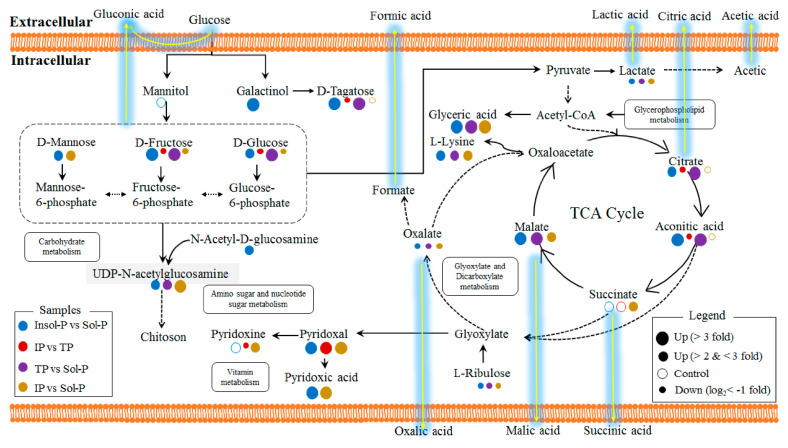
Intracellular metabolic pathways regulated by P sources in *P. oxalicum*. Identified metabolites from Q-TOF are marked with colored circles. Briefly, blue, red, purple, and yellow circles represent the metabolite intensity of the samples that are affected by insoluble P (Insol-P) vs. soluble P (Sol-P), iron phosphate (IP) vs. tricalcium phosphate (TP), TP vs. Sol-P, IP vs. Sol-P, respectively. The diameters of the cycles represent the intensities of the intracellular metabolites in four comparisons. The solid circles represent the significant increase (FC > 2 fold, *p* < 0.05) and decrease (log_2_(FC) < −1, *p* < 0.05) intensities of important metabolites comparing to each control treatments. The hollow circles represent no significant changes (0.5 < FC < 2, *p* < 0.05) of the metabolic intensities. Solid lines show the intracellular metabolic pathways that exist in *P. oxalicum*, while the dotted lines show the supposed metabolic pathways. The yellow gleam lines represent the supposed extracellular acid production pathways collecting with intracellular metabolic pathways.

**Figure 6 metabolites-10-00441-f006:**
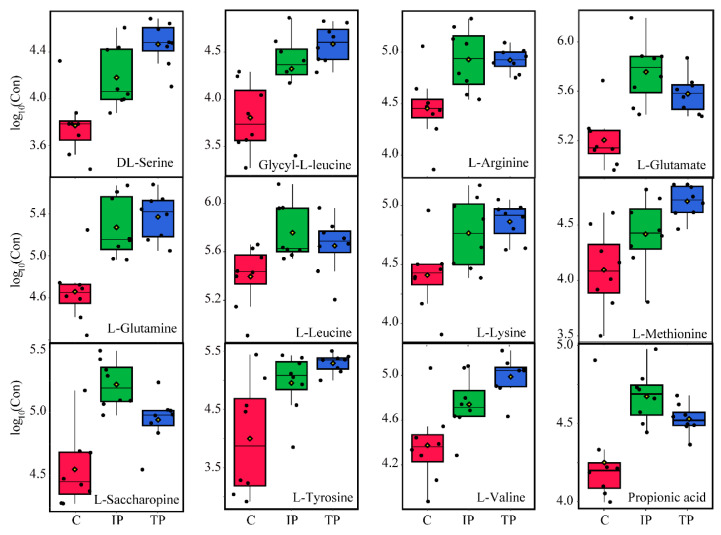
Changes in concentration distribution of important amino acid metabolites in *P. oxalicum* with tricalcium phosphate (TP-Blue), iron phosphate (IP-Green), and dipotassium phosphate (C-Red) treatments. The black dots represent the concentration of amino acids from eight samples.

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
