# Peer review of "Intracellular Metabolomics Switching Alters Extracellular Acid Production and Insoluble Phosphate Solubilization Behavior in Penicillium oxalicum"

_metabolites, 2020, doi:10.3390/metabo10110441_

Round 1
Reviewer 1 Report
The authors studied the intracellular metabolic processes of how microbes solubilize insoluble phosphorus to increase bio-available P. The paper was well, conclusions were clearly presented with strong evidence. The use of untargeted metablomics approach is highly appreciated in showing the broader picture of changes observed in different treatment.
Limitations of the study may include:
- Microbial growth were not monitored throughout the observation period.
- Since based on the result of single strain, conclusions may need to be made thoroughly. Solubilizing mechanism found in this paper could not be generalized to all fungus or all PS microorganisms. Therefore selection of multiple highly representative strains are recommended for future studies.
Author Response
Response to Reviewer 1:
Comment 1. Microbial growth were not monitored throughout the observation period.
Response: Thank for reviewer’s good suggestion to perfect the manuscript. In this study, we did not choose the microbial growth as a parameter of experimental observation, because the morphology and appearance of Penicillium oxalicum were not changed significantly during the 2-week culture process. On the contrary, we further analyzed the effects of different insoluble phosphate on extracellular acids and intracellular metabolomics of Penicillium oxalicum by discussing the changes of microbial biomass, Psol and pH during culture period.
Comment 2. Since based on the result of single strain, conclusions may need to be made thoroughly. Solubilizing mechanism found in this paper could not be generalized to all fungus or all PS microorganisms. Therefore selection of multiple highly representative strains are recommended for future studies.
Response: Thank for reviewer’s good comment to perfect the manuscript. We revised some conclusions in the manuscript to make it more thorough and specific, and added the sentence “For comprehensive and specific P solubilizing mechanism, multiple highly representative strains are needed for further studies.” in line 389-399. It is true that a single strain sometimes can not reflect the change of all fungi, but the research need follow the law from simple to complex. We see Penicillium oxalicum PSF-4 as the origin for the study of P solubilizing mechanism. In the future, we will choose more and more representative strains as the object of study, so as to make the conclusions more convincing. Thank you very much for perfecting our manuscript.
Reviewer 2 Report
I found the article “Intracellular metabolomics switching alters extracellular acid production and insoluble phosphate solubilization behavior in Penicillium oxalicum” to be an interesting approach to give in-depth information about the metabolic processes of how microbes solubilize insoluble phosphorus to increase bio-available P. I found the experimental design to be well-organized, the approach and the methods used very novel, and the whole dataset, presented either within the text or as supplementary material, to be very interesting. The text is well-written; however, some minor linguistic corrections are required. I do believe that this article can be published in Metabolites journal as soon as the authors address some minor points
- Please try to use italics whenever you are writing a species name. In the title and in some other cases in the text, it must be corrected
- Line 45. Write correctly the name of the authors and add et al. after the name. The same stands for all other cases where you just use only the first author, while you have multiple authors (e.g. Line 58 Zhao…)
- Line 77. This is the first time that you mention the terms Psol and C but you have not previously explain their meaning. Since your M&M section is at the end you must do it either here or in the last paragraph of the discussion
- Please try to reorder your graphs and tables in your supplementary materials. In-Line 85 you start with Fig. S3A. Since this is your first graph it should be S1A. Change the rest accordingly
- Lines 77-86. The way this paragraph is written is very complicated. I had to read this part several times to understand it. Please improve it.
- 1 B. How do the authors explain the low Psol values on day 9?
- The resolution of figure 2 is very low. I could hardly see what is written there. Please provide another figure
- In figure 2 the authors should not use so many times these color legends for the description of the graphs and at the same time present the same information below the graphs
- The resolution of the heat map is very low
- In Table S1 please replace “f.values and p.values” with “F-values and P-values”
- In all tables and figure legends, all abbreviations must be explained. Every one of them must be independent of the rest of the text
Author Response
Response to Reviewer 2:
Comment 1. Please try to use italics whenever you are writing a species name. In the title and in some other cases in the text, it must be corrected
Response: Thank for reviewer’s good suggestion to perfect the manuscript. We have corrected the incorrect use of italics in this manuscript.
Comment 2. Line 45. Write correctly the name of the authors and add et al. after the name. The same stands for all other cases where you just use only the first author, while you have multiple authors (e.g. Line 58 Zhao…)
Response: Thank for reviewer’s good comment to perfect the manuscript. We have modified it after each quote as required in line 45 and 58.
Comment 3. Line 77. This is the first time that you mention the terms Psol and C but you have not previously explain their meaning. Since your M&M section is at the end you must do it either here or in the last paragraph of the discussion
Response: Thank for reviewer’s good comment to perfect the manuscript. We have added the necessary explanation in line 77 and 78.
Comment 4. Please try to reorder your graphs and tables in your supplementary materials. In-Line 85 you start with Fig. S3A. Since this is your first graph it should be S1A. Change the rest accordingly
Response: Thank for reviewer’s good comment to perfect the manuscript. We have made changes as required.
Comment 5. Lines 77-86. The way this paragraph is written is very complicated. I had to read this part several times to understand it. Please improve it.
Response: Thank for reviewer’s good comment to perfect the manuscript. We have adjusted the structure of this paragraph and deleted redundant words to make its logic clearer from line 77 to 83.
Comment 6. 1 B. How do the authors explain the low Psol values on day 9?
Response: Thank for reviewer’s good comment to perfect the manuscript. We have rechecked the original data, in the previous figure, we have deleted the statistical outlier data in day 9 to control the errors. However, as the Reviewer suggested, it might be confused by other readers that the low Psol values on day 9. So we added the outlier data and redrawed the figure 1 as followed, and replaced the previous figure 1 in the manuscript. Thank you very much for perfecting our manuscript.
Comment 7. The resolution of figure 2 is very low. I could hardly see what is written there. Please provide another figure
Response: Thank for reviewer’s good comment to perfect the manuscript. We have adjusted the resolution of Fig. 2 and replaced the inappropriate figures in this manuscript.
Comment 8. In figure 2 the authors should not use so many times these color legends for the description of the graphs and at the same time present the same information below the graphs
Response: Thank for reviewer’s good comment to perfect the manuscript. In Fig. 2, we have adjusted the use of color and minimized the addition of new colors to reduce the difficulty of reading. The Fig. 2 used many colors to distinguish different low-molecular-weight organic acids (LMWOAs), but the same LMWOAs are represented by the same color. Behind the seemingly complex legends, there are connections and differences among the experimental groups.
Comment 9. The resolution of the heat map is very low
Response: Thank for reviewer’s good comment to perfect the manuscript. We have adjusted the resolution of the heat map to 600 dpi.
Comment 10. In Table S1 please replace “f.values and p.values” with “F-values and P-values”
Response: Thank for reviewer’s good comment to perfect the manuscript. We have made the modifications as required.
Comment 11. In all tables and figure legends, all abbreviations must be explained. Every one of them must be independent of the rest of the text
Response: Thank for reviewer’s good comment to perfect the manuscript. In order to avoid misunderstanding, we have added the whole words of each abbreviations in all the tables and figures. Thank you very much for perfecting our manuscript.
